# Circulating Cells with Macrophage-like Characteristics in Cancer: The Importance of Circulating Neoplastic-Immune Hybrid Cells in Cancer

**DOI:** 10.3390/cancers14163871

**Published:** 2022-08-11

**Authors:** Thomas L. Sutton, Ranish K. Patel, Ashley N. Anderson, Stephen G. Bowden, Riley Whalen, Nicole R. Giske, Melissa H. Wong

**Affiliations:** 1Department of Surgery, Oregon Health & Science University, Portland, OR 97239, USA; 2Department of Cell, Developmental and Cancer Biology, Oregon Health & Science University, Portland, OR 97201, USA; 3Department of Neurological Surgery, Oregon Health & Science University, Portland, OR 97239, USA; 4Knight Cancer Institute, Oregon Health & Science University, Portland, OR 97201, USA

**Keywords:** fusion, fusion hybrid, CAML, CHC, liquid biopsy, macrophage-tumor fusion, CTC

## Abstract

**Simple Summary:**

In cancer, disseminated neoplastic cells circulating in blood are a source of tumor DNA, RNA, and protein, which can be harnessed to diagnose, monitor, and better understand the biology of the tumor from which they are derived. Historically, circulating tumor cells (CTCs) have dominated this field of study. While CTCs are shed directly into circulation from a primary tumor, they remain relatively rare, particularly in early stages of disease, and thus are difficult to utilize as a reliable cancer biomarker. Neoplastic-immune hybrid cells represent a novel subpopulation of circulating cells that are more reliably attainable as compared to their CTC counterparts. Here, we review two recently identified circulating cell populations in cancer—cancer-associated macrophage-like cells and circulating hybrid cells—and discuss the future impact for the exciting area of disseminated hybrid cells.

**Abstract:**

Cancer remains a significant cause of mortality in developed countries, due in part to difficulties in early detection, understanding disease biology, and assessing treatment response. If effectively harnessed, circulating biomarkers promise to fulfill these needs through non-invasive “liquid” biopsy. While tumors disseminate genetic material and cellular debris into circulation, identifying clinically relevant information from these analytes has proven difficult. In contrast, cell-based circulating biomarkers have multiple advantages, including a source for tumor DNA and protein, and as a cellular reflection of the evolving tumor. While circulating tumor cells (CTCs) have dominated the circulating cell biomarker field, their clinical utility beyond that of prognostication has remained elusive, due to their rarity. Recently, two novel populations of circulating tumor-immune hybrid cells in cancer have been characterized: cancer-associated macrophage-like cells (CAMLs) and circulating hybrid cells (CHCs). CAMLs are macrophage-like cells containing phagocytosed tumor material, while CHCs can result from cell fusion between cancer and immune cells and play a role in the metastatic cascade. Both are detected in higher numbers than CTCs in peripheral blood and demonstrate utility in prognostication and assessing treatment response. Additionally, both cell populations are heterogeneous in their genetic, transcriptomic, and proteomic signatures, and thus have the potential to inform on heterogeneity within tumors. Herein, we review the advances in this exciting field.

## 1. Cancer Biomarkers: In Search of the Holy Grail

Cancer is the most common cause of death and is responsible for ~10 million deaths annually, worldwide [1]. Each disease site or histologic variant brings unique challenges in diagnosis, prognostication, and treatment. Current modalities to track cancer evolution or response to therapy include procedure-based, imaging-based, and biopsy-based technologies. Procedure-based biomarkers, such as screening colonoscopy for colon cancer or endoscopy for esophageal cancer, are effective in cancer detection but are resource-intensive, require specialized staff and equipment, and are not widely accessible [2,3,4]. Imaging-based biomarkers, such as computed tomography (CT), magnetic resonance imaging (MRI), ultrasonography, and positron emission tomography (PET) are non-invasive, repeatable tools that can be readily employed for the screening, surveillance, and assessment of therapeutic response. However, the performance of imaging devices considerably varies and protocols are not standardized [5]. Furthermore, imaging is dependent on the expertise of the interpreting radiologist: secondary interpretation of imaging-based biomarkers by sub-specialized diagnostic radiologists has been shown to uncover management-changing discrepancies in 18.6% of cases [6]. Additionally, many imaging modalities are expensive and not widely accessible in resource poor environments, thus limiting their utility as a universal approach to guiding clinical cancer care [7]. Tissue biopsy is currently the gold standard biomarker for diagnosing and evaluating tumor genetic, proteomic, or transcriptomic data; however, it is invasive, painful, carries potential adverse side effects, and is subject to false-negatives from sampling or pathologic interpretation [8]. Thus, the development of biomarkers of evolving tumors that can overcome these barriers represents a critical need in the monitoring and management of cancer [9].

The ideal biomarker is easily acquirable, non-invasive, operator-independent, and broadly relevant across a variety of diseases and treatment stages. For these reasons, peripheral blood is an optimal source for cancer biomarkers, including proteins [such as carcinoembryonic antigen (CEA) and cancer antigen 19-9 (CA 19-9)], exosomes, and cell-free tumor DNA (cfDNA)—all which derive from cancer cells [10,11,12,13,14,15,16]—as well as intact disseminated tumor-associated cell populations [17,18,19,20,21,22,23,24,25,26,27,28,29]. The requisite tumor origin of cell-derived components indicates that a cell-based assay provides the greatest diversity of information reflecting the evolving tumor biology, including genomic, proteomic, transcriptomic, and epigenetic profiles. Circulating tumor cells (CTCs) were discovered and first reported in 1869, and their study has dominated the field of circulating cells in cancer [30]. However, the limitation of CTCs as a clinically useful biomarker are increasingly apparent, highlighting the need for a focus on other tumor-derived circulating cell populations. Herein, we review atypical and newly defined circulating tumor-derived cell populations in cancer, focusing on circulating macrophage-like cells (CAMLs) [31], and circulating hybrid cells (CHCs) [29], which demonstrate exciting potential for their use to provide insights into tumor progression, tumor heterogeneity, and treatment response.

## 2. Circulating Tumor Cells

CTCs were first identified in the 19th century and are still considered the quintessential circulating neoplastic cell population in cancer [30]. While an in-depth review of CTCs in cancer is outside the scope of this review, we provide a brief overview of their advantages and disadvantages, which provides useful context when discussing other cell types. Conventionally-defined CTCs are defined by their expression of tumor-specific proteins (e.g., cytokeratin (CK) or epithelial cellular adhesion molecule (EpCAM) in epithelial malignancies), and the absence of the pan-leukocyte marker, CD45 [32,33]. CTC levels correlate with poor prognoses across a wide variety of disease sites, including colorectal cancer (CRC), pancreatic ductal adenocarcinoma (PDAC), breast cancer, and prostate cancer [17,18,19,20,21,22,23,24,25,26,27,28]. With regard to prostate cancer, CTC levels outperform the prostate-specific antigen as an early marker of response to chemotherapy [34]. Unfortunately, CTCs are rare in circulation, particularly in the early stages of many cancers. This is highlighted by the fact that a threshold of ≥5 CTCs per mL of blood (containing 5 − 10 × 10^6^ nucleated cells per mL) correlates with high disease burden when used for prognostication [17,18,19,20,21,22,23,24,25,26,27,28]. As such, detection platforms for studies exploring CTCs as a clinical biomarker must utilize complex techniques to enrich for CTCs in patient blood by utilizing combinations of density, charge, size, and surface marker expression [35,36,37,38,39,40,41,42,43,44,45,46,47,48].

In spite of these limitations, CTCs are the only circulating cell population derived from cancer with commercially available assays approved for clinical use (e.g., CellSearch^®^) [49]. However, it should be noted that while CellSearch^®^ does have FDA approval to inform prognosis in metastatic CRC, breast, and prostate cancer patients, the platform’s reliance on epithelial markers and exclusion of cells with immune antigen expression fail to capture important subpopulations with prognostic value [50,51,52,53,54]. Additionally, it is important to recognize that utility in prognostication does not necessarily equate to demonstrable benefits in care or clinician decision-making. To this end, CTC assays have been investigated as an early measure of chemotherapy response, with trial protocols triggering a change in the chemotherapy regimen if CTC levels do not drop following the initiation of 1st-line therapy. Unfortunately, none of the three trials that explored CTC correlates—the SWOG S0500 trial, CirCe01 trial, and the STIC CTC trial—demonstrated CTC-guided chemotherapy to be superior to chemotherapy guided by the judgement of a clinician informed by imaging surveillance during treatment [55,56,57]. Despite these limitations, studies have turned to combining CTCs with other circulating biomarkers such as CA19-9 [50,58], or by expanding protein marker phenotypic profiling, including the expression of cyclooxygenase-2 (COX2), leucine rich repeat containing G-protein coupled receptor 5 (LGR5), or caudal-type homeobox 2 (CDX2); in CTCs in CRC [59,60,61]; or mesenchymal markers, Vimentin, or Twist in PDAC [51,53,62,63]. To date, however, no assay utilizing a combinatorial approach, or protein marker expansion profiling has been found suitable for clinical use. Further, use of the CellSearch^®^ platform beyond prognostication is considered off-label.

While CTCs are the best-studied circulating cell in cancer with hundreds of publications over several decades, limitations undoubtedly exist: (1) their rarity across disease stages inhibits their use in early detection and as a reliable “liquid tumor biopsy” for genetic, transcriptomic, or proteomic analyses on a commercial scale; (2) their hypothesized role in cancer metastasis remains unproven; (3) to date, multiple prospective studies have failed to show CTC enumeration improves clinical decision-making to affect outcomes. Due to these limitations, there is clearly space to consider alternative circulating tumor-derived cell populations in cancer as potential biomarkers of disease burden and state. Given that CTCs are defined by their lack of immune expression (i.e., CD45-negativity), we focus on the spectrum of tumor–immune cell interactions, and how these interactions can give rise to novel circulating tumor-derived cell populations with full or partial immune identity.

## 3. Tumor-Associated Macrophages, Phagocytosis, and Cancer-Associated Macrophage-Like Cells (CAMLs)

### 3.1. Tumor-Associated Macrophages

Macrophages are a multifaceted CD45-expressing immune cell population with known diverse roles in cancer biology [64,65,66,67]. Under homeostatic conditions, macrophages serve key roles in maintaining tissue protection through pro-inflammatory signaling and direct function including the phagocytosis of dead or dying cells [68]. In cancer, tissue-resident macrophages migrate toward hypoxic and necrotic tumor areas. Additionally, specific chemotactic factors released by neoplastic cells recruit peripheral monocytes (precursors of differentiated macrophages) to repopulate and augment the pool of tissue-resident macrophages [64,65,69]. Once recruited to the tumor microenvironment (TME), local cues shape the different functional and phenotypic populations [70]. The net effect of these processes is significant, resulting in tumor-associated macrophages (TAMs) composing up to 50% of total tumor mass in some cases [66,67].

Curiously, TAMs display oppositional functional phenotypes that depend on metabolite-linked crosstalk in the surrounding TME [71]. Classically activated M1 macrophages mediate several anti-tumor functions by forming reactive oxygen and nitrogen species, secreting pro-inflammatory cytokines [tissue necrosis factor-alpha, (TNF-α) interleukin-6 (IL-6), interferon-gamma (INF-γ)] to recruit tumor-killing leukocytes, or directly phagocytosing tumor cells [72]. In contrast, alternatively activated M2 macrophages are described as tumor-promoting, breaking down the basement membrane to facilitate tumor invasion, promoting angiogenesis, and protecting against T-cell mediated anti-tumor immune responses. TAMs are now known to not belong to just two populations, but rather exist on a dynamic spectrum; however, there may still be two predominant net functions (tumor-killing and promoting) [73,74,75,76,77]. TAM populations confer prognostic significance accordingly: an increase in M1-like TAMs is a relatively favorable prognostic factor, while M2-like TAM elevation is a biomarker of poor outcomes [78]. The spectrum of states that TAMs adopt—tumor-interfacing macrophages with both tumor-suppressing and tumor-promoting roles—coupled with their antigen presenting capabilities have generated great interest in circulating cells that express immune markers as a liquid biopsy for cancer [79,80].

### 3.2. Cancer-Associated Macrophage-Like Cells

Differentiated macrophages are rarely observed in peripheral blood; however, circulating macrophage-like cells that contain vesicles harboring tumor material were identified in patients with breast, pancreatic, and prostate cancer in 2014 by Adams et al. [31]. These cells were called cancer-associated macrophage-like cells (CAMLs) based on their cell surface expression of the macrophage protein CD14. CAMLs are a morphologically heterogeneous population, generally larger than CTCs, ranging from 25 to 300 µm in size. They have atypical or multiple nuclei, and they contain phagocytosed tumor protein epitopes in the cytoplasmic vesicles. Additionally, these cells may undergo homotypic fusion with other macrophages to result in their large size and frequent multinucleated phenotype: a well-described process for macrophages in chronic inflammatory diseases [81]. Notably, these cells are immune cells and cannot re-capitulate tumorigenesis like CTCs or CHCs.

CAMLs are found in esophageal, lung, liver, pancreatic, colorectal, breast, and prostate cancer [31,82,83,84,85,86,87]. Like CTCs, CAMLs are detectable in patients with advanced cancers, but have low sensitivity for detecting disease in early stages [82,84]. Similarly, epitope-detection in monocyte (EDIM) technologies measure circulating CD14^+^/CD16^+^ activated monocytes/macrophages with internalized tumor-derived proteins such as Apo10 and transketolase-like protein 1 (TKTL1) [88,89]. Such circulating cells are highly prevalent in oral cancer, PDAC, CRC, and cholangiocarcinoma, and reflect Apo10 and TKTL1 expression in primary tumors [88,89]. Due to definitional and measurement differences, it is difficult to assess whether cells detected by EDIM platforms are CAMLs, as phagocytosis is just one method by which circulating cells in cancer can reflect tumor characteristics, such as cell death associated with treatment response.

Published data indicate that while having higher levels of CAMLs pre-treatment is correlated with shorter overall and progression free survival [82,85], CAML enumeration tends to track with treatment response, where CAML numbers transiently increase in response to chemotherapy [31]. Notably, numbers of CAMLs change in an inverse relationship to CTCs (or CHCs), which decrease with therapy response, highlighting the functional nature of disseminated macrophage-like cells. Presumably these cells survey and engulf dying cancer cells, and thus increase in numbers when treatment is effective. Interestingly, CAML size differs in relation to treatment status; higher pre- and post-treatment CAML sizes in esophageal cancer and non-small cell lung carcinoma (NSCLC) have been correlated to worsened overall survival, possibly related to more aggressive disease subtypes [86,87].

## 4. Tumor Cell Fusion, Tumor-Immune Hybrids, and Circulating Hybrid Cells

### 4.1. Intratumoral Tumor–Immune Hybrids

The theory that fusion between immune cells and cancer plays a functional role in tumor progression and metastasis was first introduced by the German pathologist, Otto Aichel in 1911 [90]. This concept was predicated upon the notion that macrophage/leukocyte phenotypes expressed in neoplastic cells could be associated with functions known to drive cancer metastasis (i.e., migration, extravasation, immune evasion) [90]. While cell fusion occurs in both homeostatic and noncancerous inflammatory states [91,92,93], it has recently been described in malignancy [94,95,96,97,98,99,100,101,102]. Heterotypic cell fusion hybrids are generated in cell co-culture [29], in vitro murine models of injury-regeneration [94,96], in tumorigenesis [29,103], and in human cancer patients [29,33,104,105,106,107,108,109,110,111]. Much like the sampling of a tumor’s genome and proteome by CAMLs, tumor-immune hybrid cells harbor immune and neoplastic cell attributes, and thus provide important information with regard to tumor state and the tumor microenvironment. Given that neoplastic-immune hybrid cells harbor functional attributes of both parental cells of origin, these hybrids are implicated in influencing tumor progression and the metastatic spread of disease (Figure 1).

Spontaneous cell fusion is observed in real time by in vitro live-imaging of co-culturing murine MC-38 CRC cells expressing red fluorescent protein (RFP) and green fluorescent protein (GFP) expressing macrophages. Hybrid cells harboring cytoplasmic GFP and intact nuclear RFP are characteristic. GFP^+^/RFP^+^ hybrid cells are mitotically active, with sustained co-expression of fluorescent markers in daughter cells across multiple generations [29]. Other groups similarly characterized fusion hybrids in murine breast cancer and glioblastoma models and with in vitro human breast cancer cell lines [33,112,113,114,115]. These findings delineate cellular fusion from other immune cell functions and phenomena such as phagocytosis and trogocytosis [116].

Findings from in vitro studies suggest that cell fusion hybrids may participate in the metastatic cascade. MC38-derived hybrids display enhanced migratory and invasive properties relative to unfused MC38 cells as measured using a Boyden chamber assay [29]. In addition to the acquisition of macrophage migratory and invasive behaviors, in vitro-derived melanoma-macrophage hybrid cells initiated tumorigenesis when orthotopically injected into recipient mice. In this setting, unfused tumor cells required an order of magnitude higher numbers of cells to support tumor growth [29,33]. Similar findings have also been shown in human and murine models of ovarian, breast, and gastric carcinoma, with evidence suggesting that fusion promotes the epithelial-to-mesenchymal transition (EMT) and activates Wnt/β-catenin signaling pathways [103,117,118]. Notably, growth at the primary site of tumor–immune hybrid injection has not been shown in all studies [119]. Finally, in vitro-derived MC38 hybrid cells generated pulmonary metastases with higher numbers and growth than unfused MC38 cells using an experimental model of metastasis [29].

A number of groups have demonstrated the spontaneous generation of cell fusion hybrids using in vivo models. Inflammation and epithelial proliferation are key mediators within the TME required to facilitate cellular fusion, and these conditions can be recreated in murine models by utilizing γ-irradiation and bone marrow derived cell (BMDC) transplantation [94,96]. Using this model system, several BMDC populations displayed the ability to fuse with host intestinal epithelia, including common myeloid and lymphoid progenitors, mature B/T cells, and macrophages. However, macrophages displayed the most robust cell fusion capacity, at a proportion significantly higher than other BMDC lineages [95]. This suggests that the macrophage is one principal fusogenic leukocyte. Further, given the importance of the monocyte/macrophage lineage within the TME, this indicates that circulating tumor-immune cells could provide information on the evolving immune landscape within the tumor.

Hybrid cells generated in bone marrow transplanted mice with intestinal tumors harbored transcriptomic signatures of both macrophage and epithelial cells, while simultaneously displaying a unique transcriptomic signature [95]. Several genes associated with metastatic spread were upregulated in hybrid cells relative to unfused tumor cells, including activated leukocyte cell adhesion molecule (ALCAM), runt-related transcription factor 1 (RUNX1), and fms-related tyrosine kinase 4 (FLT4) [95,120,121,122]. Similar findings were identified within in vitro models of spontaneous fusion hybrid formation in sarcoma [123]. The theme of cellular reprogramming following fusion has been recapitulated across numerous studies in cancer and other physiologic states, including injury-regeneration in multiple organ sites [92,94,96,115,124,125,126,127,128,129,130,131,132,133].

Tumor–immune hybrids in circulation harbor features of solid malignancies. The strongest in vivo evidence comes from female recipients of sex-mismatched bone marrow transplants who subsequently developed solid organ malignancies. In these patients who developed PDAC, tumor specimens harbored cells positive for Y chromosomes and tumor markers, such as CK, indicating cellular fusion between male donor leukocytes and recipient tissue intratumorally [29]. The list of malignancies in which tumor–immune hybrids have been identified continues to expand, now including PDAC, renal cell carcinoma, head and neck squamous cell carcinoma (HNSCC), lung adenocarcinoma, NSCLC, melanoma, prostate adenocarcinoma, and ovarian adenocarcinoma [29,103,109,110,115,134,135,136,137].

Together, the available evidence on macrophage–tumor cell fusion hybrids suggests that fusion imparts a highly proliferative, migratory, and tumorigenic phenotype relative to unfused cancer cells, one that is the result of genetic and phenotypic reprogramming and provides a basic mechanism for tumor–immune cell fusions to participate in cancer progression and metastasis, with a corresponding potential utility as a biomarker.

### 4.2. Tumor–Immune Hybrids in Circulation

Tumor–immune hybrids disseminate from the primary tumor into peripheral blood where they are termed circulating hybrid cells (CHCs). These cells were first identified in a murine model of tumorigenesis, leveraging the co-expression of tumor and macrophage proteins [29]. RFP-labeled B16F10 melanoma cells, subdermally injected in Actin-GFP mice, facilitated the detection of RFP^+^GFP^+^ CHCs in peripheral blood using flow cytometry and confirmed by FACS-sorting and downstream analyses [29]. Notably, CHCs comprised 90% of the disseminated tumor cells detected in peripheral blood. Unfused tumor cells (RFP^+^GFP), analogous to CTCs, made up the remaining minor population of circulating tumor-derived cells [29,33,104]. Importantly, CHCs are heterogeneous with respect to their immune and tumor antigens [29,33]. However, the vast majority of murine RFP^+^GFP^+^ CHCs expressed the pan-leukocyte antigen CD45; thus, CD45-expression can be used as a marker for hybrid cell identity in human cancer patients to detect CHCs [29]. Given their abundance and phenotypic similarities with invasive and migratory intratumoral fusion hybrids, CHCs have been implicated in the metastatic cascade, but may also serve an important function as a plentiful source of tumor-derived cells for analysis.

Detection of CHCs using CD45 or other immune marker expressions has been reported in a wide array of disease sites, including PDAC; HNSCC; uveal melanoma; gastric, breast, colon, rectal, lung, gastrointestinal stromal tumors; and glioblastoma [29,33,50,106,107,111,138,139,140,141,142,143,144]. The earliest report of human CHCs was in breast cancer patients in 2014, where CK^+^/CD45^+^circulating cells were associated with poor survival [139]. Subsequently, Toyoshima et al. compared the tumor initiation capacity of isolated EpCAM^+^/CD45^+^ and EpCAM^+^/CD45^−^ circulating cells from patients with gastric cancer. The injection of these fractions into immunodeficient mice revealed the enhanced tumorigenicity of the CD45^+^ population compared to the CD45^−^ population: greater numbers of mice injected with EpCAM^+^/CD45^+^ cells developed tumors [138]. Similarly, from melanoma patients, Clawson et al. successfully cultured peripheral blood cells that co-expressed epithelial (CK, EpCAM), melanocyte (Melan-A, ALCAM), and macrophage (CD204, CD206, CD163) proteins, and demonstrated their downstream tumorigenic capacity when injected into immunodeficient mice [106]. Critically, these cells were identified in the primary tumors of melanoma patients, to support the primary tumor origin of CHCs.

The link between tumor and circulating hybrids is perhaps best demonstrated by the identification of the Y chromosome in CHCs (CK^+^/CD45^+^ and tumor-specific MUC4) in female recipients of sex-mismatched BMT who subsequently developed cancers [29]. Dietz et al. further demonstrated primary tumor and CHC relationships, as evidenced by the identification of Kirsten rat sarcoma virus (KRAS) mutation in a subset of CHCs isolated from the blood of a patient with PDAC [33]. They went on to present conserved protein expression patterns between tumor biopsy and disseminated CHCs from breast cancer patients undergoing therapy [33]. Importantly, CHCs have also been shown to harbor tumor copy number alterations [145]. Together, these data highlight CHC detectability, specificity to the tumor of origin, and conserved ubiquity across myriad cancer types, indicating their considerable potential as a circulating tumor-derived biomarker with the potential to non-invasively inform tumor biology.

### 4.3. CHCs as a Biomarker in Human Malignancy

Tumor heterogeneity evolves with time and in response to therapeutic treatment; thus, the extent to which phenotypic heterogeneity of CHCs reflect the evolving tumor biology is of great interest. Recently, Dietz and colleagues profiled this heterogeneity within tissue biopsies using cyclic immunofluorescence (cyCIF) via antibodies specific to stromal, immune, epithelial, and vascular compartments of the tumor [33]. In two patients with breast cancer, phenotypic changes within neoplastic cells that occurred during systemic treatment were characterized relative to a pre-treatment biopsy. The post-therapy biopsy demonstrated changes in proliferative, epithelial, hormonal, and stem marker-expressing cells in response to treatment. Leveraging cyCIF technology, they demonstrated that phenotypes observed in the primary tumor were reflected in the disseminated CHCs, indicating that tumor hybrids and CHCs share heterogeneous phenotypes [33]. Given the alignment of protein expression in CHCs with the primary tumor, utilizing CHCs to characterize the disease phenotype and surveil disease progression during treatment may be on the horizon. Serial blood draws may reveal phenotypic changes in CHCs, such as the maintenance of stem features that relate to the evolving tumor environment [33]. Further, this liquid biopsy approach could identify treatment-resistant populations within the tumor allowing for better directed, cell-specific treatment. Along with the single-cell analyses of CHCs isolated from tumors, the heterogeneity of CHCs isolated from peripheral blood may suggest the clinical relevance of targetable antigens and warrants further investigation.

While the term CHC is novel, atypical circulating cells with macrophage/monocyte-like characteristics in patients with cancer have been widely reported, supporting the potential use of this population as a biomarker [50,138,139,140,141,142,146,147,148,149,150]. Indeed, initial investigations of CHCs highlight potential through their correlation with disease stage, survival, and response to treatment. CHC numbers in untreated patients with PDAC correlate with disease stage and survival, and with metastasis in lung cancer [29,111,151]. Manjunath et al. corroborated these findings in NSCLC patients, finding that the co-positive “tumor-macrophage fusion cell” population correlated with the tumor stage. Further, they demonstrated that the number of larger, so-called “giant” CHCs independently predicted survival [111]. Additionally, CHC enumeration showed promise in the pre-operative setting for oral cavity HNSCC by predicting the presence of both clinically overt and occult nodal metastases [105]. Strikingly, CHC levels following neoadjuvant therapy for either rectal or esophageal adenocarcinoma adequately discriminated a pathologic complete response from an incomplete response and were associated with the recurrence risk in esophageal adenocarcinoma [104]. Two of these patients with rectal cancer were serially monitored during neoadjuvant therapy, with the CHC number decreasing in response to therapy and increasing before the clinical evidence of disease progression. Collectively, these studies show considerable promise for CHCs as a biomarker of disease status, treatment response, and prognosis.

## 5. Discussion

A cellular-level analysis provides genomic, epigenetic, and transcriptomic data of the tumor’s biology and the extent of heterogeneity therein. This information is critical to the diagnosis, treatment, and management of cancer patients. While procurement of the tumor tissue through invasive biopsy or surgery is currently the gold standard, non-invasive approaches such as monitoring atypical circulating tumor-associated cells provide an exciting area of ongoing research and possibilities to guide decision-making. Circulating cells in cancer can be leveraged in a multitude of ways, as they can yield DNA, RNA, protein, and vesicles for use in clinical assays, including those of cell-free DNA (cfDNA) if intentionally lysed [152]. Unlike cfDNA, however, monitoring intact circulating cells has the potential advantage of monitoring tumor heterogeneity and its evolution in real time, through an analysis of phenotypic subpopulations of disseminated cells. While a recent longitudinal study of circulating tumor DNA (ctDNA) in patients with liver-metastatic CRC demonstrated that changes in levels during systemic therapy were highly prognostic, by itself this provides little actionable information as to the underlying resistant disease mechanisms [153]. In contrast, monitoring the evolution of circulating cellular subpopulations may provide a readout of response/resistance to treatment, such as the proportion of circulating cancer-derived cells in breast cancer that express or contain the estrogen receptor (ER) or overexpress human epidermal growth factor-receptor 2 (HER2) in breast cancer, both of which are therapeutic targets. Similarly, monitoring the population of triple negative circulating cells in a patient with ER^+^ or HER2^+^ breast cancer in targeted therapy may allow early identification of the emerging resistant disease. Additionally, for CHCs it is yet to be determined whether they are direct effectors of metastatic risk; however, longitudinal monitoring of CHCs or other circulating effector cell populations may have clinical utility in predicting the metastatic risk at diagnosis and after curative-intent procedures or therapeutics [104,105].

Disseminated neoplastic cells in peripheral blood have complementary attributes that can inform on tumor biology. While CTCs are the most thoroughly studied circulating cells in cancer, it is acknowledged that their use is limited by their exceedingly low numbers. This has hampered their robust characterization, as well as the gap between their identities for clinical use versus their evolving phenotypes in research. Further prospective studies to date have not shown their utility in guiding treatment decision-making. In contrast, circulating cell populations expressing immune proteins such as CHCs and CAMLs are detected in higher numbers [29,33,104], and demonstrate exciting promise for early detection, diagnosis, and surveillance of a wide range of cancers. One limitation is that these entities have not yet been studied prospectively. CAMLs are a direct result of anti-tumor immune activity within the tumor, and they may be best suited to measuring cancer cells that are responding to treatment; however, no evidence exists demonstrating that CAMLs can identify treatment-resistant populations. Indeed, one would expect treatment-resistant disease to undergo phagocytosis less frequently than treatment-sensitive disease, and CAMLs are unlikely to proportionally reflect the heterogeneity of the tumor. In contrast, cell fusion results in heterogeneous cell fusion hybrid progeny, even from monoclonal parental populations. This may allow the monitoring of tumor–immune hybrids and CHCs to detect sufficient phenotypic variability that aligns with primary or metastatic tumor heterogeneity. CHCs appear to adopt a range of tumor phenotypes, are not dependent on tumor cell death for their formation, and may more proportionally reflect tumor heterogeneity: all distinguishing features from CAMLs. To date, however, there have been few publications investigating both of these populations compared to the immense volume of CTC-centric research spanning the last several decades, with even fewer publications evaluating their biology.

A mechanistic and prospective study of these intriguing, newly described cell populations is required to determine how they may be harnessed as clinically useful biomarkers. Not only are CHCs and CAMLs generally more plentiful than CTCs [29,33,104], allowing higher signal-to-noise ratios when comparing diseased versus healthy states, but they also contain relevant tumor proteomic, transcriptomic, and genomic data. Additionally, combining an analysis of multiple circulating cellular biomarkers may have benefits compared to using single analytes for analyses. For example, a ratio of CAMLs to CHC/CTCs or their clinically-relevant subpopulations may reflect how successful the immune system is at controlling the tumor, with higher ratios favoring a tumor-killing state. Interventions to enhance immune surveillance or promote tumor cell death may influence such a ratio through reducing the number of disseminated tumor cells.

## 6. Conclusions

Circulating neoplastic-immune hybrid cells represent a novel subpopulation of circulating neoplastic cells derived from the primary tumors of cancer patients. Unlike CTCs, both CAMLs and CHCs are readily attainable across different disease types and stages, and thus are better suited to be utilized as cancer biomarkers. To date, both CAML and CHC levels have been demonstrated to correlate with survival and treatment outcomes, and treatment-related protein expression pattern changes in primary tumors have been shown to reflect in their CHC counterparts. Many questions exist in this exciting field as neoplastic-immune hybrid cells have extraordinary potential to transform the care and management of disease in cancer patients.

## Figures and Tables

**Figure 1 cancers-14-03871-f001:**
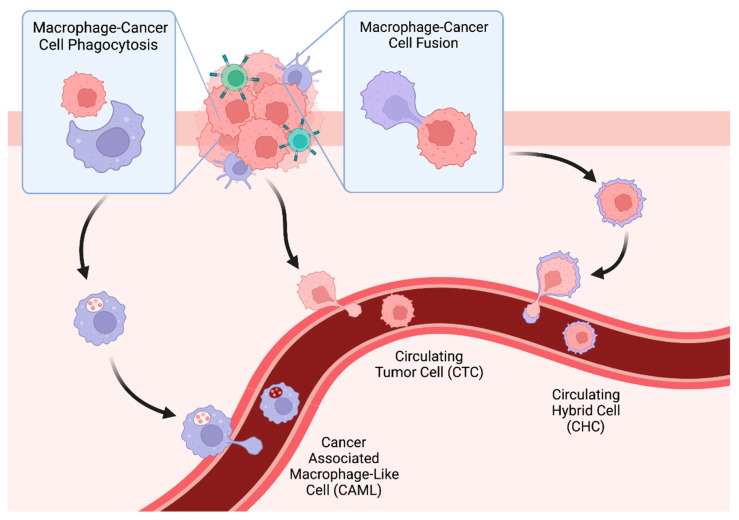
Circulating hybrid cells. CAMLs (immune cells), CTCs, and CHCs (neoplastic cells) are disseminated into peripheral blood from the tumor. Monitoring their presence in blood provides different information into the tumor evolution or response to therapy.

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
