# Peer review of "Circulating Cells with Macrophage-like Characteristics in Cancer: The Importance of Circulating Neoplastic-Immune Hybrid Cells in Cancer"

_cancers, 2022, doi:10.3390/cancers14163871_

Round 1

Reviewer 1 Report

Dear Authors,

Good review of the literature. However, have you looked into age-related immune profiling of cells? How does the immune phenotype change with age? Could age-related immune profiling of cells be used in the diagnosis of disease?

Author Response

Thank you for this important question. This is certainly an interesting concept that we and others have yet to explore. In our studies we have not stratified our analyses on discrete groups related to age, but have interest in investigating early onset of CRC in younger cohorts, so this could facilitate asking age-related an immune profiling questions.

Reviewer 2 Report

This is an important and timely review article. Especially the authors have written in a concise approach. The manuscript is well-written and it almost could be published in the current version. I recommend its publication after the minor modification.

1. For "Current modalities to track cancer evolution or response to therapy include procedure-based, imaging-based, and biopsy-based tech-nologies. " here it is better to cite the related literatures to support it. For a case, the "image-based" might be referred to Journal of colloid and interface science, 2021, 549, 493-501, where NIR fluorescence images-guided therapy was done.

2. Is there full name for COX2? If there is, it is better to give the full name.

Author Response

1. For "Current modalities to track cancer evolution or response to therapy include procedure-based, imaging-based, and biopsy-based tech-nologies. " here it is better to cite the related literatures to support it. For a case, the "image-based" might be referred to Journal of colloid and interface science, 2021, 549, 493-501, where NIR fluorescence images-guided therapy was done.

  1. Thank you for bringing this to our attention. While, our intention with this section is to bring light to the fact that currently cancer evolution and response to therapy is generally evaluated with procedures (i.e. surgeries, colonoscopy, etc.), imaging (CT, ultrasound, MRI, etc.), and/or biopsies (fine needle aspiration, core needle biopsies, etc.) which we highlight in the subsequent sentences, your point is well-taken. We now include the reference listed in your feedback that provides an extremely interesting study utilizing a phototherapeutic nanoplatform for NIR laser triggered tumor therapy in in-vitro and in-vivo animal models.

2. Is there full name for COX2? If there is, it is better to give the full name.

2. The full name for COX-2 is cycloxogenase-2, and we have updated our manuscript to clarify this acronym, as well as other acronyms as they appear in the manuscript.

Reviewer 3 Report

The review-manuscript of Sutton et al is well written and the topic of circulating neoplastic-immune hybrid cells is a burning issue in oncology. I have some reflection that I would like to bring to authors attention.

Authors should accentuate the gap between the "conventional" definition of CTC which is used to guide clinical trials and the characteristics of these cells founded by recent research, one of all is epithelial to mesenchymal transition. This divergency is one of the main problems of studying CTCs in clinical trials, authors should stress that this “oversight” should not be repeated for neoplastic-immune hybrid cells.

For this reason, I suggest authors to implement the manuscript with the known biological aspects and the role of these hybrid cells in the metastatic cascade.

Author Response

The review-manuscript of Sutton et al is well written and the topic of circulating neoplastic-immune hybrid cells is a burning issue in oncology. I have some reflection that I would like to bring to authors attention.

Authors should accentuate the gap between the "conventional" definition of CTC which is used to guide clinical trials and the characteristics of these cells founded by recent research, one of all is epithelial to mesenchymal transition. This divergency is one of the main problems of studying CTCs in clinical trials, authors should stress that this “oversight” should not be repeated for neoplastic-immune hybrid cells.

For this reason, I suggest authors to implement the manuscript with the known biological aspects and the role of these hybrid cells in the metastatic cascade.

 This is an important point. We now include this sentiment in the discussion.